# Transcriptomic landscape and chromatin accessibility uncover pivotal regulators driving programmed larval-larval molting in the domesticated silkworm

Yun Wang[1], Xin Yang[1], Junfeng Hong[2], Lingyi Li[1], Xia Ling[2], Liang Qiao[2]*, Ze Zhang[1]*, Wei Sun [1]*

**1** Laboratory of Evolutionary and Functional Genomics, School of Life Sciences, Chongqing University, Chongqing, China, **2** Chongqing Key Laboratory of Vector Control and Utilization, Institute of Entomology and Molecular Biology, College of Life Sciences, Chongqing Normal University, Chongqing, China

\* qiaoliangswu@163.com (LQ); zezhang@cqu.edu.cn (ZZ); sunwei077@cqu.edu.cn (WS)

## Abstract

Insects undergo periodic ecdysis to shed their old chitinous exoskeleton and form a new cuticular layer. The steroid hormone 20-hydroxyecdysone (20E) is widely recognized as a central regulator of insect molting. Acting as a signaling molecule, 20E pulses orchestrate gene expression in a concentration- and time-dependent fashion. However, investigations into the transcriptomic and epigenomic alterations linked to dynamic 20E fluctuations remain limited. In this study, we explored the temporal dynamics of epidermal transcriptomes and genome-wide chromatin accessibility during the larval-larval molting cycle of the silkworm, *Bombyx mori*. Our results unveiled pronounced shifts in gene expression and chromatin architecture between early and late molting stages, correlating with ascending and descending 20E titers, respectively. Chromatin footprint analysis identified the Ecdysone receptor (EcR) and Grainy head (GRH) as early-stage regulators. Strikingly, during late molting phases, we uncovered a novel regulatory axis involving CCAAT/enhancer-binding protein (C/EBP) alongside the established factor Fushi-tarazu f1 (βFTZ-F1). Moreover, decline of the 20E titer triggers the expression of C/EBP, which subsequently regulates *βFtz-f1* expression through promoter binding. Furthermore, epidermal-specific knockout of *C/EBP* and *βFtz-f1* genes led to dysregulation of cuticular protein and chitin biosynthesis genes, impairing new cuticle formation. Collectively, our multi-omics dissection illuminates the dynamic regulatory circuitry coordinating epidermal remodeling and establishes a hierarchical transcriptional cascade governing cuticular renewal. These findings advance our understanding of hormone-driven developmental transitions in insects.

**Data availability statement:** The raw and processed RNA-seq and ATAC-seq data generated in this study is depositing to the National Genomics Data Center under BioProject accession code PRJCA032352.

**Funding:** This work was supported by the National Natural Science Foundation of China (32070499 to W.S., 31772527 to L. Q.), the Scientific and Technological Research Program of Chongqing Municipal Education Commission (KJZD-K202200507 to L. Q). The funders had no role in study design, data collection and analysis, decision to publish, or preparation of the manuscript.

**Competing interests:** The authors have declared that no competing interests exist.

## Author summary

Insect molting is a precisely timed biological process where shedding of the rigid exoskeleton enables growth. This process is orchestrated by pulses of the steroid hormone 20-hydroxyecdysone (20E), which triggers sequential gene expression and cuticle remodeling. However, the full cascade of transcription factors (TFs) activated by the rise and fall of 20E, particularly those acting late in the molting cycle, remains incompletely understood. Here, we investigate how 20E coordinates larval molting in the silkworm *Bombyx mori* through time-resolved transcriptome and chromatin accessibility profiling. We reveal that dynamic changes in gene expression are tightly coupled to fluctuations in chromatin openness during molting stages. Computational footprinting uncovers key transcription factors governing early and late molting events, including a previously unrecognized 20E-responsive regulator. Remarkably, we demonstrate that this novel factor is essential for activating the late-stage gene βFtz-f1 and ensuring successful larval-larval molting. Our findings provide new mechanistic insights into how hormone pulses control developmental timing through coordinated chromatin and transcriptional reprogramming.

## Introduction

How animals coordinate development and differentiation within defined temporal frameworks are fundamental questions in developmental biology. Insect model systems provide an optimal platform for investigating the molecular mechanisms that govern progressively developmental transitions in multicellular organisms, primarily due to the distinct and clearly-defined molting and metamorphosis stages within their individual developmental processes [1]. These transition events are tightly regulated by endocrine hormones which is believed to be conserved during the growth and sexual maturation of mammals [2].

The cuticular exoskeleton of insects plays essential roles in supporting the animal body, protecting the internal organs and preventing water loss [3]. The exoskeleton primarily consists of the epicuticle and procuticle layers; the latter is further divided into exocuticle and endocuticle, which are composed of cross-linked structural cuticular proteins (CPs) and the rigid polysaccharide chitin [4,5]. To accommodate growth, insects undergo periodic molting to overcome the rigid constraints of their exoskeletons. This molting process is driven by a series of intricately arranged events, including the degradation of the old cuticle, formation of a new cuticle, and ecdysis. Various external and internal stimuli can initiate this molting process, among which the steroid hormone ecdysteroid is considered one of the most critical regulators [6].

In insects, the major active ecdysteroid is known as 20-hydroxyecdysone (20E). Previous studies have demonstrated that pulses of this hormone function as temporal signals to coordinate molting events [7,8]. During the pre-molt period, the hormone titer in hemolymph increases rapidly, inducing detachment of the epidermis from the

old cuticle (apolysis), followed by a peak [6]. Around the hormone peak, the outer epicuticle is deposited [9]. After apolysis, the ecdysteroid titer declines while the new epi- and pro-cuticles are secreted [8]. Cuticular protein genes are the main components of the cuticle, and the timing of their expression is thought to dictate the formation of distinct exoskeleton layers [8,10–13]. Previous studies have shown a number of CPs were regulated by 20E in a concentration-dependent manner. Several CPs could be directly induced by low doses of 20E, whereas the transcript of bulks of the genes requires the declining titers after the hormone peak [14,15]. When the exogenous 20E was injected into the animals at the time of endogenous 20E peak, the newly formed cuticle was delayed and expression of some CPs were also repressed [8,16]. Therefore, the start and end points of the ecdysteroid pulse at the proper developmental stages are essential to control the normal molting process.

Fluctuations of the hormone titer sequentially activate a number of transcription factors (TFs) which further regulate the progressively molting events [17,18]. The ecdysone-signaling cascade initiates with the upregulation of early genes, such as Broad, ecdysone inducible gene E74, E75 and E93, in response to low levels of the hormone, followed by the induction of delayed early genes (hormone receptor 3 and hormone receptor 4) at high hormone concentrations [12,17,19]. After the decline of the 20E titer, late genes, such as Fushi-tarazu f1 (βFtz-f1), start to express. It should be noted that some early responsive regulators have been identified and functionally dissected in various insects, whereas few late factors were studied for better understanding the essential roles of the hormone pulse during molting and metamorphosis processes [17].

In this study, we performed a time-course analysis of epidermal transcriptomes during the larval-larval molting process in the silkworm, and profiled the dynamic changes in gene expression along with the fluctuation of 20E concentration. Integrative analysis of open chromatin profiling and the transcriptome landscape revealed the changes in gene expression were highly accompanied by the chromatin accessibility. Subsequently, the computational footprints of transcription factors uncovered key regulators in the early and late molting stages, respectively. Remarkably, we identified a previously unrecognized 20E-responsive TF that plays essential roles in regulating *βFtz-f1* expression and larval-larval molting events. Our findings illuminate the sequential changes in gene expression during molting stages and provide new insights into the 20E signaling pathway.

## Results

### Morphological changes of the integument during the larval molting process

To observe the morphological changes of the integument during the larval molting process, we performed hematoxylin and eosin (H&E) and chitin staining on larval integuments. Six time points, identified using the spiracular apolysis index in previous study, were selected to represent the main larval molting events, encompassing an ecdysone pulse (Fig 1A) [6]. At the feeding stage (stage A), when the concentration of 20-hydroxyecdysone (20E) was low, the epidermal cells were tightly adhered to the procuticle, forming a relatively complete structure. Chitin in the procuticle layer was also found to adhere to the epidermal cells (Figs 1B and S1). As the concentration of 20E began to rise, the larvae entered the early molting stage (stages B and C1), during which apolysis of the old cuticle from the epidermal cells was observed. Holes appeared in the procuticle, likely due to the digestion of the old layer. At the middle molting stage (stage D3), the concentration of molting hormone peaked, significantly enlarging the apolytic space between the old cuticle and the epidermis. During the late molting period (stages E1 and E2), a newly thick layer of chitin formed outside the epidermal cells, indicating the onset of new epidermis formation. Consistent with the previous study [6], the morphology of the larval integument undergoes significant changes before and after mid-molting stage corresponding to the increasing and decreasing of 20E titer, respectively.

### Dynamic transcriptome in epidermis during larval molting process

We then performed time-series transcriptome to study the dynamic landscape of gene expression throughout the silkworm larval molting process (Fig 1A). Differential expression analysis indicated that most genes exhibited significant changes

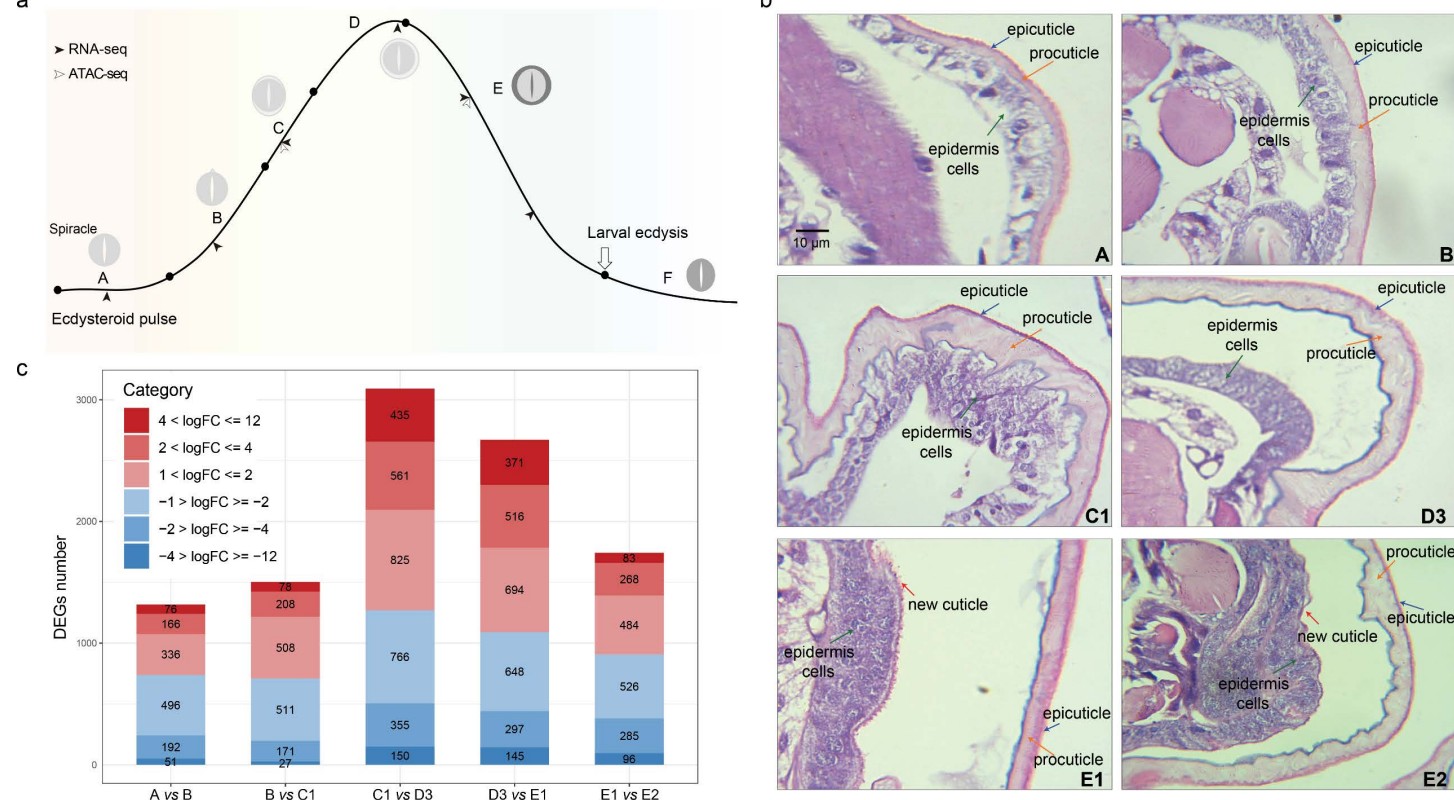

**Fig 1. Overview of transcriptome landscape of epidermis during silkworm larval-larval molting process. (a)** Experimental design. Schematic outline of RNA and ATAC sequencing and time points of sample collection. The letters (A to F) embedded in the (a) means the different molting stages according to the spiracle index reported by Kiguchi and Agui 1981. The main appearance changes of spiracle for staging the larval moult. The black line represents developmental changes in hemolymph ecdysteroid concentration based on the previous data [6]. **(b)** Morphological changes of the integument stained by H&E staining during the larval molting process. Epidermis were dissected from the 4th instar larvae at different molting stages according to the spiracle features described in **(a)**. The hematoxylin stains the nuclei blue purple, and eosin stains the cytoplasm and other structures pink. **(c)** Distribution of number of differentially expressed genes (DEGs) by log2 fold change (logFC) values. The DEGs that are up (blue) or down (red) between two adjacent molting stages were identified by DESeq2 (|logFC| > 1, adjusted P value ≤ 0.05). The numbers in the bar chart represent the count of DEGs within the corresponding logFC value range.

primarily at the two transition points: from stages C1 to D3 and D3 to E1 among all comparisons between adjacent stages (Fig 1C). Therefore, the molecular analysis underscores the D3 stage as a major transition point, distinguishing the effect of increasing and decreasing hormone titers, which correlates with the substantial morphological changes observed between early and late stages (Fig 1B).

Furthermore, we investigated developmental scenarios for gene expression throughout larval molting, focusing on the clusters which contain genes with typical expression pattern at specific developmental points (Fig 2A) [20]. Genes in cluster 1 were preferentially transcribed at the feeding stage, and Gene Ontology (GO) enrichment analysis suggested that they were involved in metabolic related activity, such as oxidoreductase activity, fatty acid synthesis activity and carbohydrate metabolic process (Fig 2B). Cluster 2 and cluster 3 represented genes with high expression during the early-molting process (B and C1), and the enriched GO terms, such as hydrolase activity, peptidase activity and transferase activity, were related to degradation of old cuticle. Notably, several chitinases and peptidases involved in the degradation of chitin and proteins exhibited increased transcript levels during stages B and C1 (S2A Fig) [21]. The expression profiles of those

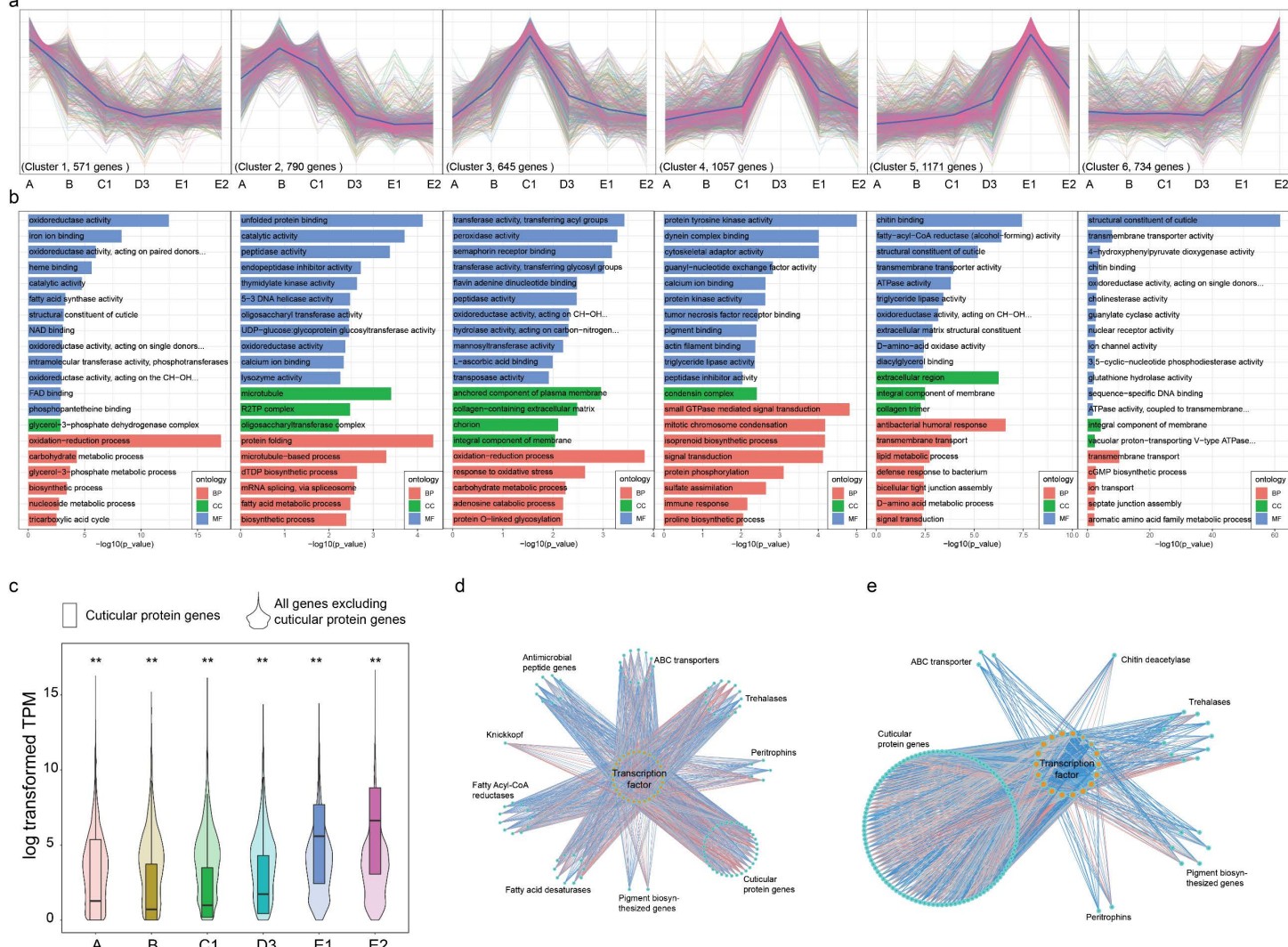

**Fig 2. Transcriptome profiling of silkworm epidermis identifies stage specific DEG modules. (a)** Clustering of DEGs using maSigPro into 6 nonredundant groups. The genes in one cluster mean that those genes have the similar expression pattern. (b) 20 top-score GO terms enriched in each cluster. Adjusted $P$-value ≤ 0.05. BP means biology process; CC means cellular component; MF means molecular function. **(c)** Comparison of transcript abundances between cuticular protein genes (box plot) and all genes excluding cuticular protein genes (violin plot), ** means Mann-Whitney U Test, $P < 0.01$. **(d)** Gene co-expression network of transcription factors and their related genes in the cluster 5. **(e)** Gene co-expression network of transcription factors and their related genes in the cluster 6. For (d) and **(e)**, the DEGs involved in the 5th cluster and 6th cluster identified in (a) were subjected to construct co-expression network by WGCNA, respectively. Then the subnetworks containing transcription factors (orange circles) and their associated genes (green circles) were extracted and constructed.

enzyme genes correlated with increasing 20E titer, indicating that they are the 20E-inducible genes [22–24]. Additionally, cluster 4 consisted of genes predominantly expressed at the middle molting stage (D3), coinciding with peak 20E levels. Functional analysis showed these genes were enriched in terms related to cell signaling transduction (e.g., signal transduction, protein kinase activity, and tyrosine activity). A total of 1,905 genes (1,171 in cluster 5 and 734 in cluster 6) were typically upregulated during the late molting stages (E1 and E2). Furthermore, the genes in these two clusters shared several GO terms, including structural constituent of cuticle, chitin binding, and transmembrane transport, contributing to the biosynthesis of new cuticular proteins and chitin (S2A and S2C Fig). Cluster 5 contained 11 genes classified under the

"fatty-acyl-CoA reductase (alcohol-forming) activity" category, which may be involved in the synthesis of cuticular hydrocarbons (S2B Fig) [25]. In summary, the distinct clustering of genes at specific stages may aid in identifying those involved in the morphological and physiological changes during the larval molting process.

As the main components of the insect epidermal layer, we identified 207 CP genes with discernable expression signals (Transcripts Per Million, TPM > 1), which can be categorized into six groups based on conserved amino acid sequence motifs [26]. We then investigated whether genes belonging to the same group exhibit co-expression, and found all types of cuticular proteins, except the RR-3 type, tended to express together at E1 and E2 stage when the larva starts to form new cuticle layer (S2D Fig). Furthermore, cuticular protein genes displayed significantly higher transcript abundances than the remaining genes in the entire genome at the E1 and E2 stages (Mann-Whitney U Test, $P < 0.01$) (Fig 2C). The marked transcript intensity and temporally biased expression patterns of cuticular genes during the late molting process corroborate the process of synthesizing new cuticle. Additionally, 28 CPs showed higher transcript abundances during the larval feeding stage, potentially contributing to the mechanical properties of the exoskeleton [27]. In summary, our data emphasize the expression timing of cuticular protein genes at proper stages is an important factor to maintain the cuticle structure.

Given that most cuticular protein and chitin biosynthesis genes are predominantly expressed during the late molting stages, it is of interest to identify regulatory factors involved in this process. We constructed a co-expression network of transcription factors (TFs) involved in clusters 5 (Fig 2D) and [6] (Fig 2E) using the Weighted Gene Co-Expression Network Analysis (WGCNA) strategy to survey correlations between pairs of genes [28]. Cluster 5 contained 29 core TFs, such as silk-gland factor 3 (sgs3) and Chorion specific CCAAT/enhancer-binding protein (C/EBP), which co-expressed with 33 cuticular proteins, 11 fatty acid reductases and 13 ABC transporters etc. For cluster 6, 18 core TFs, such as βFtz-f1, were identified and linked with 109 cuticular proteins and 6 pigment pathway genes etc. The co-expression analysis suggests that these TFs could play important roles during the late molting process.

## Hormonal regulation of the molting process

The increase and decline of ecdysteroid levels at proper developmental time points are essential for the normal progression of molting events [8]. Importantly, the expression patterns of cuticular related genes, such as cuticular protein genes and chitin metabolic genes, coordinate with the hormone fluctuation (Fig 2). Therefore, we firstly examined the transcription factors involving in 20E signaling pathway by transcriptome. During the silkworm larval molting process, genes controlled by the 20E switch on and off in a specific sequence as the hormone levels rise and fall (S3 Fig). First, the ecdysone receptor (EcR) and its partner ultraspiracle (USP) turn on as 20E starts to increase. This activates early genes like ecdysone-induced protein 78 (E78) and hormone receptor 3 (Hr3), which peak quickly. Genes like E75 and Hr4 closely follow the hormone's ups and downs. Later, when hormone levels peak, E74 turns on and stays active. Finally, as the hormone drops, late genes like Hr38 and βFtz-f1 activate. Collectively, these results reconfirmed a conserved transcriptional pattern of genes involved in the ecdysone hierarchy as in *Manduca sexta* in response to hormone fluctuations during larval molting stages [17].

To study how the end of ecdysone pulse affects the molting process at a molecular level, we injected exogenous 20E into hemolymph of the 4th instar larvae at D3 stage to delay the decline of the hormone concentration (S4 Fig). Consistent with previous study in different insects, the artificially elevated 20E level could significantly prolong the completion of ecdysis for over 24 hours [8,16]. Morphological changes of the integument showed exogenous 20E could inhibit biosynthesis of the new cuticle layer at 6 and 12 hours after treatment (hAT) (Fig 3A). Transcriptomic analysis also demonstrated that 20E treatment could significantly repressed the expression of genes enriched in "structural constituent cuticle" and "chitin binding" terms (Fig 3B). Interestingly, the expression changes of cuticular protein genes following 20E treatment mirrored their transcription patterns during the larval molting process (Fig 3C). The cuticular protein genes expressed during the early molting stages (cluster 3 and 4, labeled with yellow branch) were upregulated after hormone treatment, while most

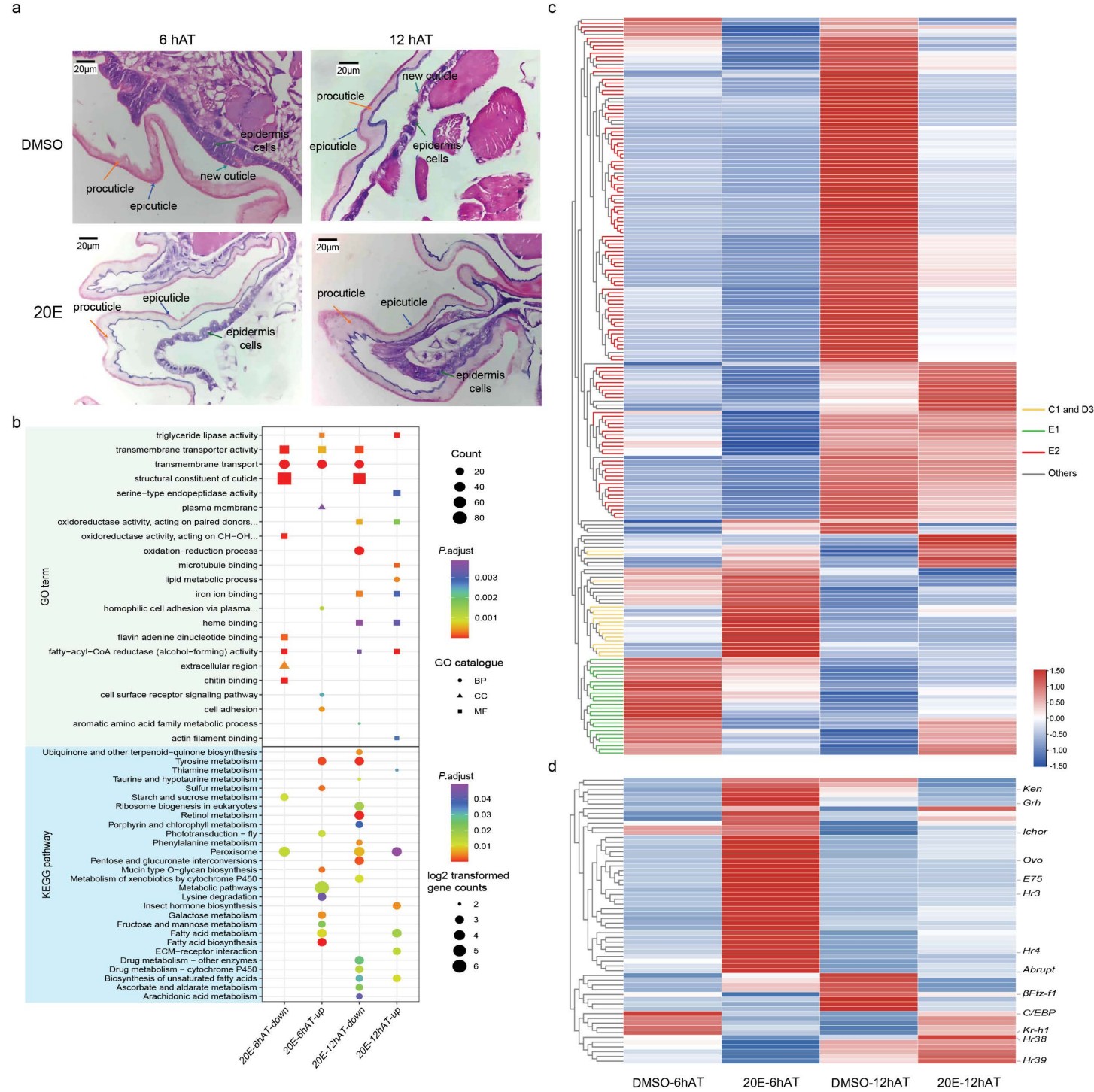

**Fig 3. Transcriptome profiling of silkworm epidermis treated with exogenous 20E. (a)** Morphological changes of the integument after exogenous 20E treatment. Exogenous 20E or DMSO were injected into the haemolymph of the 4th instar larvae at D3 stage. Then the integument of the treated larvae was dissected to perform H&E staining. hAT means hours after treatment. **(b)** GO and KEGG terms enriched in DEGs after 20E treatment. **(c)** Heatmap of the expression of differentially expressed cuticular protein genes (CPs). Yellow branches represent CPs predominantly expressed at C1 or D3 stages; Green branches represent CPs predominantly expressed E1 stage; Red branches represent CPs predominantly expressed E2 stage. **(d)** Heatmap of the expression of differentially expressed transcription factors.

of cuticular protein genes belonging to cluster 6 (red branch) were inhibited at 6 and 12 hAT. Genes in cluster 5 (green branch) were primarily reduced at 6 hAT with high concentration of 20E but were induced at 12 hAT when the hormone fell to low level. Those data indicated that the pulse of 20E is required for the expression of those cuticular protein genes. To identify potential factors involved in 20E signaling cascade, we examined the expression changes of transcription factors and found that 60 TFs were significantly influenced by the exogenous 20E treatment (Fig 3D). Among them, 68.3% (41/60) were upregulated at 6 hours after injection, including known hormone-inducible TFs such as grainy head (*Grh*), *E75*, *Hr3*, and *Hr4*. Conversely, *βFtz-f1*, a late response TF, was repressed by high 20E levels at both time points. Thus, our results suggest that the decline of 20E at appropriate developmental stages is essential for maintaining the signaling network and completing the larval molting process.

## Identification of critical transcription factors regulating molting process

We then performed Assay for Transposase-Accessible Chromatin sequencing (ATAC-seq) analysis to explore genome-wide chromatin accessibility and associated accessible TF-binding sites. Epidermis from two time points (Fig 1A, C1 and E1), representing early and late molting stages, was selected for ATAC-seq and RNA-seq. The majority of ATAC-seq reads were enriched at promoter regions overlapping the transcription start sites (TSS) (S5A Fig). By comparing the ATAC-seq data between the two stages, we detected 2577 and 2894 open peaks in the C1 and E1 samples, respectively (Fig 4A). Furthermore, 767 genes around C1 open peaks and 955 genes around E1 open peaks were identified. Joint analysis of chromatin accessibility and transcriptome showed that genes with high expression levels are distributed around the open chromatin regions at the same developmental stages ($r=0.73$, $P<2.2e-16$) (Fig 4B). Finally, 296 (C1-up-open) and 510 (E1-up-open) genes with higher expression intensity and open chromatin status were identified in C1 and E1 stages, respectively. GO enrichment showed that C1-up-open genes were annotated as proteolysis, serine-type endopeptidase activity and catalytic activity (Figs 4C, 4E and S5B), whereas E1-up-open genes were enriched in chitin binding, structural constituent of cuticle and transmembrane transport *etc* (Figs 4D, 4F and S5B).

To further identify transcription factors controlling the dynamic expression changes, we analyzed the DNA sequences within C1 and E1 accessible chromatin regions to find differentially enriched motifs. Compared to the ratio of background occurrence, 24 and 25 DNA binding sites were significantly enriched in C1 and E1 open chromatin regions, respectively (S6A and S6B Fig). Among them, ECR/USP and Grainy head (GRH) motifs in C1 were the two most prominently enriched (Figs 5A, 5B and S6A). In the E1 accessible chromatin regions, C/EBP and βFtz-f1 binding sites were the top 2 enriched motifs (Figs 5C, 5D and S6B). Interestingly, C/EBP and βFtz-f1 were also identified as core regulators in our co-expression network analysis (Fig 2D and 2E). We then focused on these four transcription factors and identified target genes with one of the predicted TF motifs in the open chromatin regions. The expression profiles of the transcription factors were positively correlated with those of their target genes (Figs 5E, S6C, S6D, S6E and S6F). Moreover, GO enrichment analysis indicated that GRH and ECR/USP targeted genes were associated with catabolism-related biology process including serine-type endopeptidase activity, proteolysis and catalytic activity, suggesting the two TFs could regulate the degradation of epidermal proteins and chitin (Fig 5F). For the two TFs identified in E1 stage, their targeted genes were enriched in new epidermal synthesis process, such as structural constituents of cuticle, chitin binding, lipid biosynthetic processes and transport activity. Interestingly, we detected C/EBP binding sites in the promoter region of *βFtz-f1*, as well as a βFtz-f1 motif in the promoter region of *Cyp18a1*, a key enzyme involved in 20E inactivation (Fig 5G and 5H). We further validated C/EBP and βFtz-f1 could bind to some of their targeted genes by Chromatin Immunoprecipitation (ChIP) (S6G and S6H Fig).

## Functional analysis of *C/EBP* and *βFtz-f1*

Since the decline of 20E levels is crucial for new cuticle formation, we mainly focused on the functions of two core regulators (*C/EBP* and *βFtz-f1*), during the late larval molting process. The two genes started to transcript from E1 stage as 20E levels decreased, indicating that reduced the hormone concentration is required for their expression. To prove this

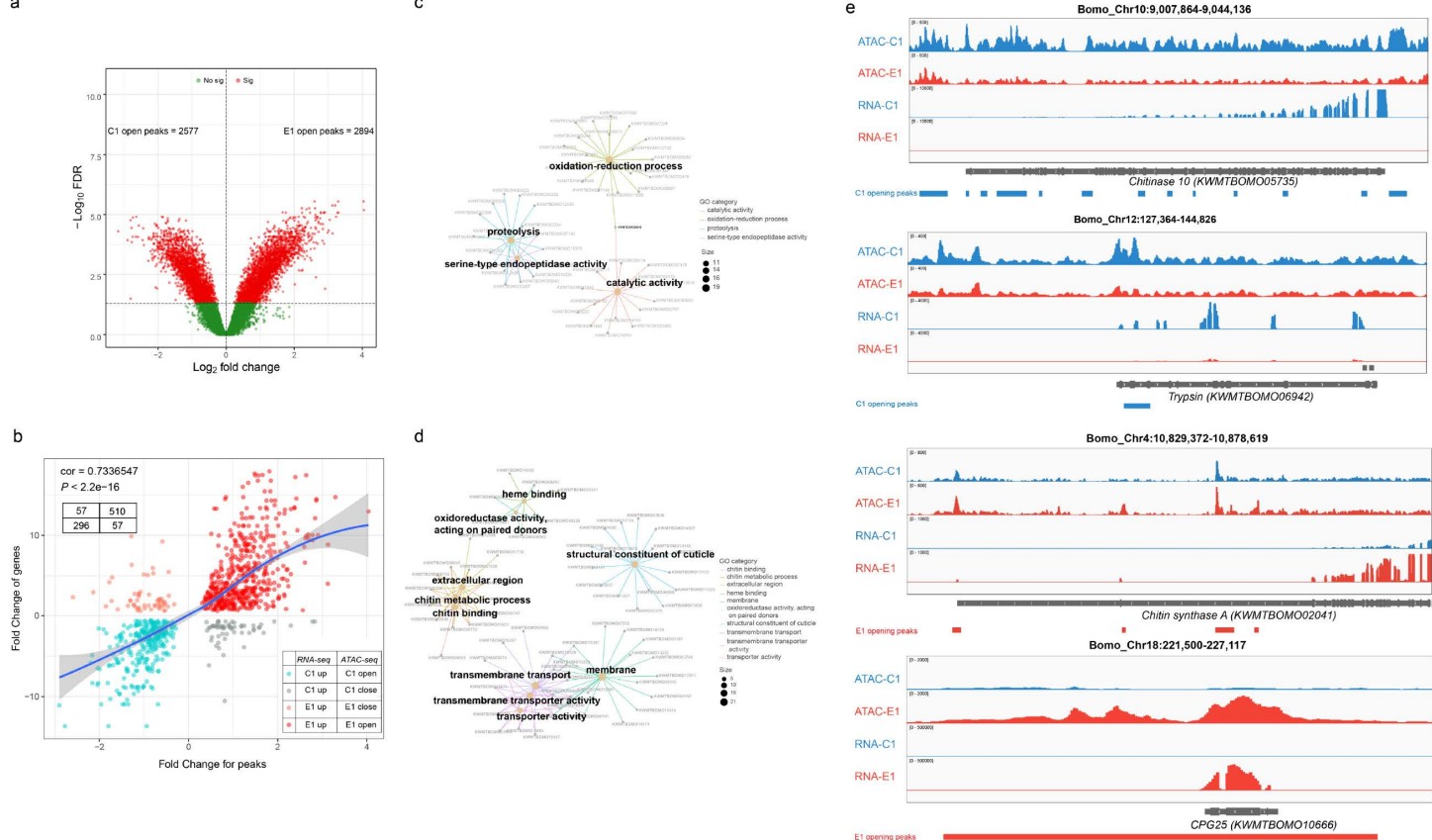

**Fig 4. Chromatin accessible changes of silkworm epidermis between C1 and E1 molting stages. (a)** Volcano plot for differential accessibility peaks between C1 and E1 molting stages, adjusted *P* value <0.05, change folds greater than 2. **(b)** Correlation of differentially accessibility chromatin and differentially expressed genes between C1 and E1 molting stages (Pearson r = 0.73, *P* < 2.2e-16). **(c)** GO terms enrichment for genes with higher expression intensity and open chromatin status at E1 stage. **(d)** GO terms enrichment for genes with higher expression intensity and open chromatin status at C1 stage. **(e)** Genome browser shot of normalized ATAC and RNA sequencing counts. ATAC-seq and RNA-seq data from C1 stage in blue and data from E1 stage in red. Height indicates normalized ATAC-seq or RNA-seq signal.

regulatory manner, we injected 20E into hemolymph of the 4th instar larvae at D3 stage, when the endogenous hormone starts to decline. The exogenous 20E could elevate the hormone concentration, which was then rapidly degraded. Nine hours after treatment (hAT), the hormone titers decreased to a level similar to that in 3 hours after D3 stage of control larvae (S4 Fig). Thus, the exogenous 20E injection could monitor the fluctuation of the hormone during molting process. Compared with the DMSO treatment, the artificially elevated hormone could significantly repress the expression of *C/EBP* and *βFtz-f1* gene at 3 and 6 hours after injection, when the 20E was still high. Then, *C/EBP* started to highly express at 9 hours after 20E treatment, while its transcription levels fell to low in the control at the same time (Fig 6A). The *βFtz-f1* gene also showed increased transcription levels at 9 hours after 20E treatment, however, the transcription intensity was still significantly lower than that of the control (Fig 6B). In comparison, we injected 20E mimic (methoxyfenozide) which could not be degraded by silkworm, to continually activate the hormone signaling pathway [29]. Methoxyfenozide largely reduced the transcript of *C/EBP* and *βFtz-f1* gene to low level at any tested time points. In summary, the decline of 20E at proper molting stage is essential to sequentially induce the expression of *C/EBP* and *βFtz-f1*.

We further explored the functions of *C/EBP* and *βFtz-f1* gene by CRSPR/Cas9 system (S7A Fig). In a previous study, we established a conditional knockout system by expressing *Cas9* gene with an epidermis specific promoter

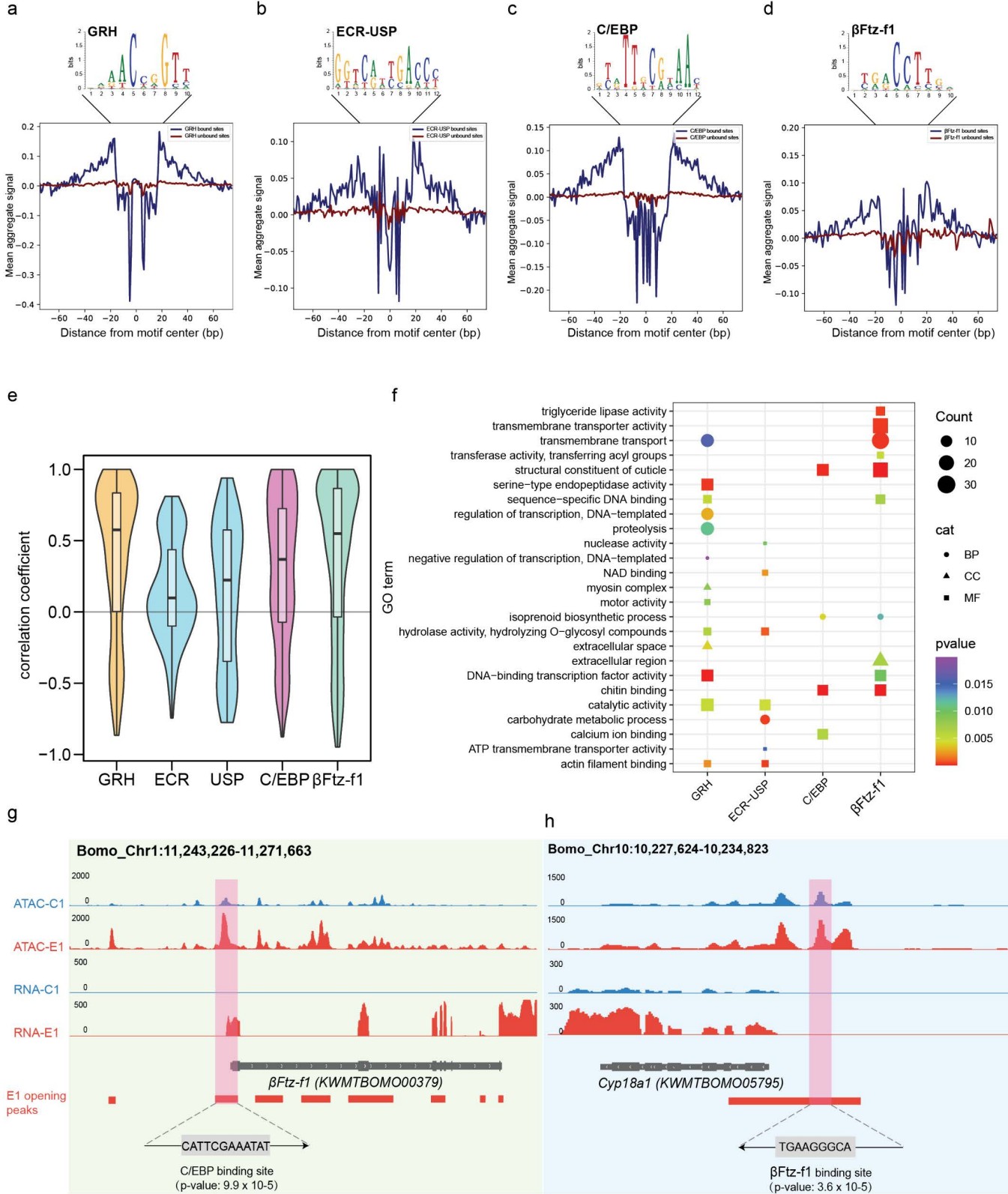

**Fig 5. Identification of key transcription factors at early (C1) and late (E1) molting stages.** Aggregate ATAC-seq footprint plot for GRH **(a)**, ECR-USP **(b)**, C/EBP (c) and βFtz-f1 (d) obtained over binding sites within the genome. For (a) and **(b)**, the open chromatin regions at C1 stage were

used to do the motif enrichment. Then the footprinting analysis were performed and visualized to detect the characteristic "footprint" pattern (depletion of cleavage events in the bound region) around the motif instances. **(e)** Correlation coefficient between variation in gene expression of the indicating transcription factors and their potential targeted genes. **(f)** GO terms enrichment for the targeted genes of the transcription factors. **(g)** Genome browser shot of normalized ATAC and RNA sequencing counts around *βFtz-f1* gene. The C/EBP binding site is highlighted in pink. **(h)** Genome browser shot of normalized ATAC and RNA sequencing counts around *Cyp18a1* gene. The βFtz-f1 binding site is highlighted in pink.

(*G25p-Cas9*) [30]. Furthermore, we examined the temporal expression of *Cas9* gene in the transgenic silkworm, and found the high nuclease gene expression in late embryos. During larval molting stage, its expression started at E1 stage and reached the high level at E2 stage (S7B Fig). By crossing *G25p-Cas9* transgenic silkworm (with green fluorescent marker) with ubiquitous U6 promoter driven *C/EBP* or *βFtz-f1* sgRNA transgenic silkworm (with red fluorescent marker), we could disrupt the function of the interested genes only in epidermal cells (S7A Fig). Subsequently, we performed amplicon sequencing to analyze the mutation frequency at the target sites of the larvae with double fluorescents. The average mutation rates in the *C/EBP* sgRNA1 and sgRNA2 targeted regions were 55.2% and 21.4%, respectively (S7C and S7D Fig). The intermediate mutation frequency indicates that the conditional knockout system generated a mosaic *C/EBP* mutant phenotype. For the *βFtz-f1* gene, 97.33% of sgRNA1-targeted sites were mutated, whereas sgRNA2 failed to induce mutations (S7C and S7D Fig). Nevertheless, our data demonstrate that *βFtz-f1* was successfully mutated in nearly all larvae.

We then examine the phenotypes of the mutations. None of the *G25p>C/EBP-KO* or *G25p>βFtz-f1-KO* larvae could develop into the final instar. Moreover, approximately 60% *G25p>C/EBP-KO* larvae died during larval molting stages (Fig 6C). Those larvae failed to shed the old cuticle, leaving the old layer covering the newly formed cuticle, and died eventually during molting stage. The remaining larvae died during feeding stages. More importantly, these larvae shrank into small body size, indicating their body wall might not prevent water loss (Fig 6A). To support this hypothesis, we reared *G25p>C/EBP-KO* larvae under high humidity, which significantly elevated the survival rate of the mutants (Fig 6C). Nevertheless, all larvae died before the final instar under this condition. Furthermore, an Eosin Y penetrate test showed the dye could easily enter bodies of *G25p>C/EBP-KO* larval, whereas almost no Eosin Y permeated the body walls of wild-type (*G25p>+*) and *G25p>βFtz-f1-KO* larvae (Fig 6D). Moreover, integument of *C/EBP* mutant showed significantly thinner procuticle layer and less chitin content (Fig 6E and 6F). In the case of *G25p>βFtz-f1-KO* animals, over 90% larvae failed to shed the old cuticle and died during the larval molting stages (Fig 6A and 6C). Similar to the *C/EBP* mutant phenotype, *βFtz-f1* mutant also exhibited thinner procuticle layer and less chitin content (Fig 6E and 6F). Strikingly, chitin fibres in procuticle layer of wild-type larvae were arranged in parallel to each other to form chitin sheet, whereas the chitin fibres in *βFtz-f1* mutant were randomly organized, potentially affecting the structure and function of the cuticle (Fig 6E). Overall, both *C/EBP* and *βFtz-f1* are essential to the formation of new cuticle and the completion of the molting process.

We then performed RNA-seq to survey how the two TFs regulate molting process at a molecular level. Compared to wild-type larvae, 892 genes were down-regulated and 1172 genes were up-regulated in the epidermis of the *C/EBP* mutant. Next, we combined these DEGs with C/EBP binding genes from ATAC-seq data and C/EBP co-expression genes from time-series RNA-seq data. Among them, 55.6% down-regulated genes (496) in *C/EBP* mutant had C/EBP binding site and/or co-expressed with *C/EBP* gene, whereas only 3.07% up-regulated genes (36) could overlap with them (Fig 7A). The results indicated C/EBP predominantly acts as a positive regulator of its targets. Interestingly, GO analysis indicated the overlay genes (496) were significantly enriched in cuticular protein and wax biosynthesis, supporting the loss of protection function of body wall from *C/EBP* mutant (Figs 7B, S8A and S8B). Besides, we also found disruption of *C/EBP* could repress the expression of *βFtz-f1*, confirming their regulatory relationship (S8C Fig). For *βFtz-f1*, 650 down-regulated genes and 860 up-regulated genes were identified. Twenty-five percent (164) of down-regulated genes overlapped with βFtz-f1 binding genes and/or βFtz-f1 co-expression genes (Fig 7C). Those genes primarily function in

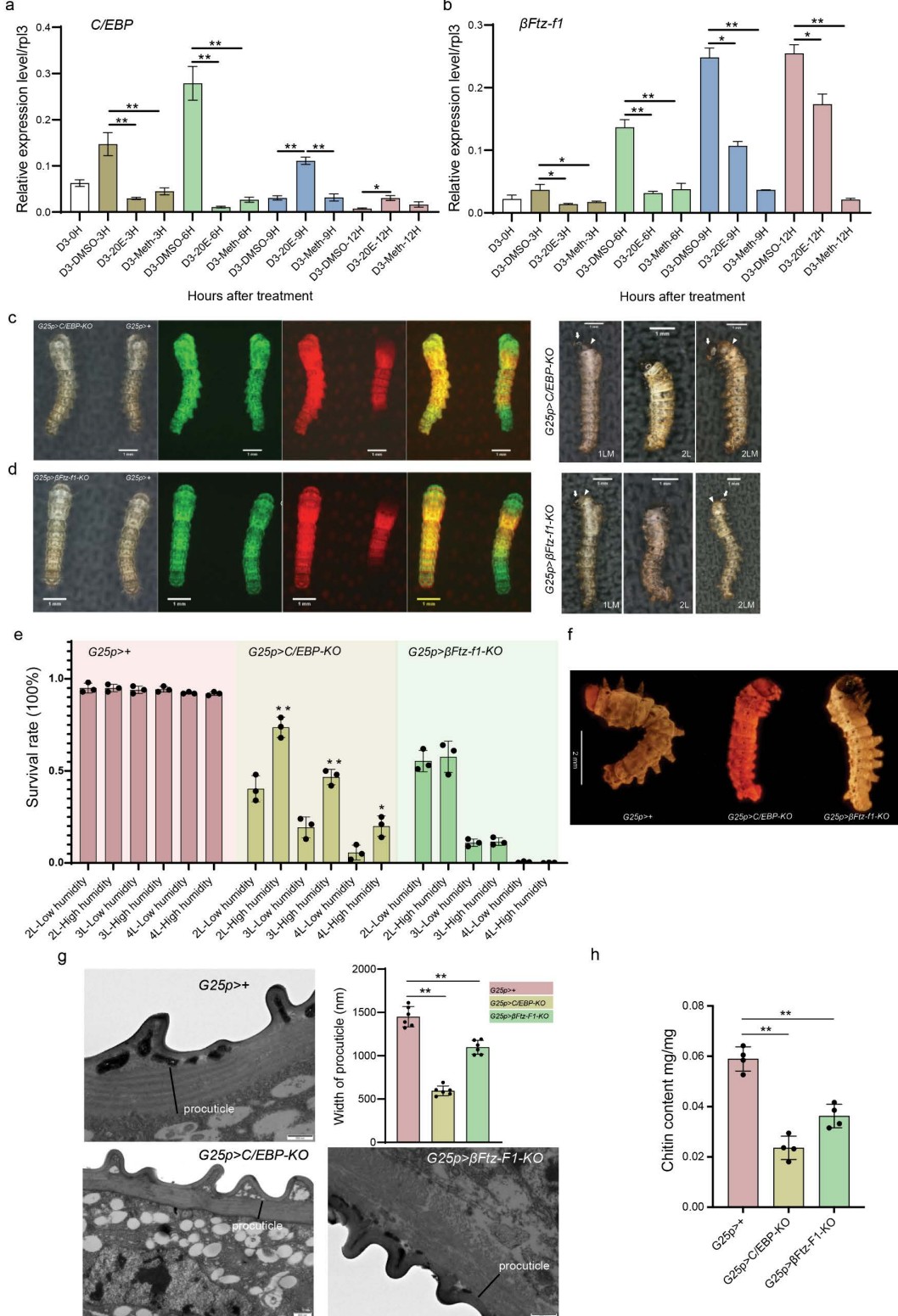

**Fig 6. Function of C/EBP and βFtz-f1 gene during the silkworm larval-larval molting stages. (a)** The effect of exogenous 20E and 20E mimic (methoxyfenozide) treatment at D3 stage on the expression of *C/EBP* gene. *n* = 3 biologically replicates. *$P$ < 0.05, **$P$ < 0.01. **(b)** The effect of exogenous

20E and 20E mimic (methoxyfenozide) treatment at D3 stage on the expression of *βFtz-f1* gene. $n = 3$ biologically replicates. *$P < 0.05$, **$P < 0.01$. Phenotype of the *C/EBP* (c) and *βFtz-f1* (d) mutated larvae. For (c) and **(d)**, the green and red fluorescents in the first two rows are used to identify the transgenic silkworms. The larvae with double fluorescents are gene mutated silkworms (*G25p>gene-KO*), and the individuals with only green fluorescents are used as control (*G25p > +*). The last row indicates the larvae dying at different developmental stages. nLM means larvae died at 1st or 2nd larval molting stages. nL means larvae died during feeding stage of 1st or 2nd instar. Arrow means old head, and arrowhead represents new head. **(e)** The survival rate of mutated larvae and control cultured at different humidity. $n = 3$ biologically replicates. *$P < 0.05$, **$P < 0.01$. **(f)** Eosin Y penetrate test for newly-ecdysed 2nd instar mutated larvae and control. **(g)** The ultrastructure of integument collected from newly-ecdysed 2nd instar mutated larvae and control by TEM. The width of procuticle layer from six samples. $n = 6$ biologically replicates. **$P < 0.01$. **(h)** The chitin content of integument collected from newly-ecdysed 2nd instar mutated larvae and control. $n = 4$ biologically replicates. **$P < 0.01$.

cuticle biosynthesis (Figs 7D and S8A). Among them, we also detected two proven cuticular protein genes, *wcp2* (*KWMIBOMO13002*) and *wcp5* (*KWMIBOMO13007*), which was directly regulated by βFtz-f1 [31] (S8A Fig). Collectively, our results demonstrate βFtz-f1 and C/EBP may directly regulate the expression of cuticle related genes, and further determine the deposition of new cuticle layer.

## Discussion

Insects undergo periodic ecdysis to shed their old chitinous cuticle and form a new layer, a process precisely regulated by fluctuations in ecdysone titers. Thus, insect integument seemed to be a representative model to investigate the regulatory effect of the hormone. In this work, we first profiled the gene expression in the integument during the silkworm larval-larval molting process Consistent with previous study, rising ecdysone levels activate genes for molting enzymes (e.g., protein/chitin degraders), digesting the old cuticle. During the subsequent hormone decline, genes for new cuticle synthesis (wax, cuticular proteins, chitin) are induced. This gene expression timing tightly matches the observed formation of the new cuticle layer (Fig 2).

In insect genomes, over 100 cuticular protein (CPs) genes belonging to several distinct families have been identified [32]. Previous studies supposed that the expression timing of CPs during development is also crucial for assembling insect cuticle layer and determining the mechanical properties of the exoskeleton [33–35]. By time-series transcriptome data, we detected several stages predominantly expressed CPs. For example, 28 CPs showed higher transcript abundances during the larval feeding stage, which were thought to regulate flexibility of the larval mature cuticle. Among them, *CPR2* gene was proven to be highly expressed after larval molting. Dysfunction of the gene in the silkworm could reduce the chitin content and tensile property of the larval cuticle [27]. More strikingly, most of the CPs from various families showed high and specific expression pattern during the late molting stages (E1 or E2 stages), contributing to the formation of exocuticle and endocuticle layers along with the decline of 20E titer. High-dose 20E injection at the endogenous hormone peak markedly suppressed CP expression. Notably, exogenous 20E exerted stage-specific effects: at high levels (6 hAT), both E1- and E2-specific CPs were strongly repressed (S4 Fig). As hormone levels declined to those typical of the E1 stage in controls, E1 CP expression resumed, whereas E2 CPs remained repressed. Therefore, our results emphasize the decline of 20E could sequentially regulate different targets in a concentration-dependent manner, and this is essential to complete the normal larval molting process [36–38].

Identifying key transcription factors activated by the decline of 20E that control new cuticle formation represents a significant research objective. Our study demonstrated that both C/EBP and βFtz-f1 function as essential regulators during the late molting stage. βFtz-f1 is a well-known TF essential for regulating larval ecdysis in insects [17,18], and its knockdown often results in impaired larval or pupal molting [39–43]. βFtz-f1 also regulates ecdysone biosynthesis of ecdysone in steroidogenic gland [44,45], making it challenging to distinguish its direct transcriptional roles from indirect effects via hormone levels. To address this, we employed a conditional knockout system to specifically disrupt *βFtz-f1* function in epidermal cells. Most of the *βFtz-f1* mutated larvae could not shed the old cuticle and died during the molting stage. Moreover, the expression of chitin related genes and lots of cuticular protein genes were repressed in mutant larvae,

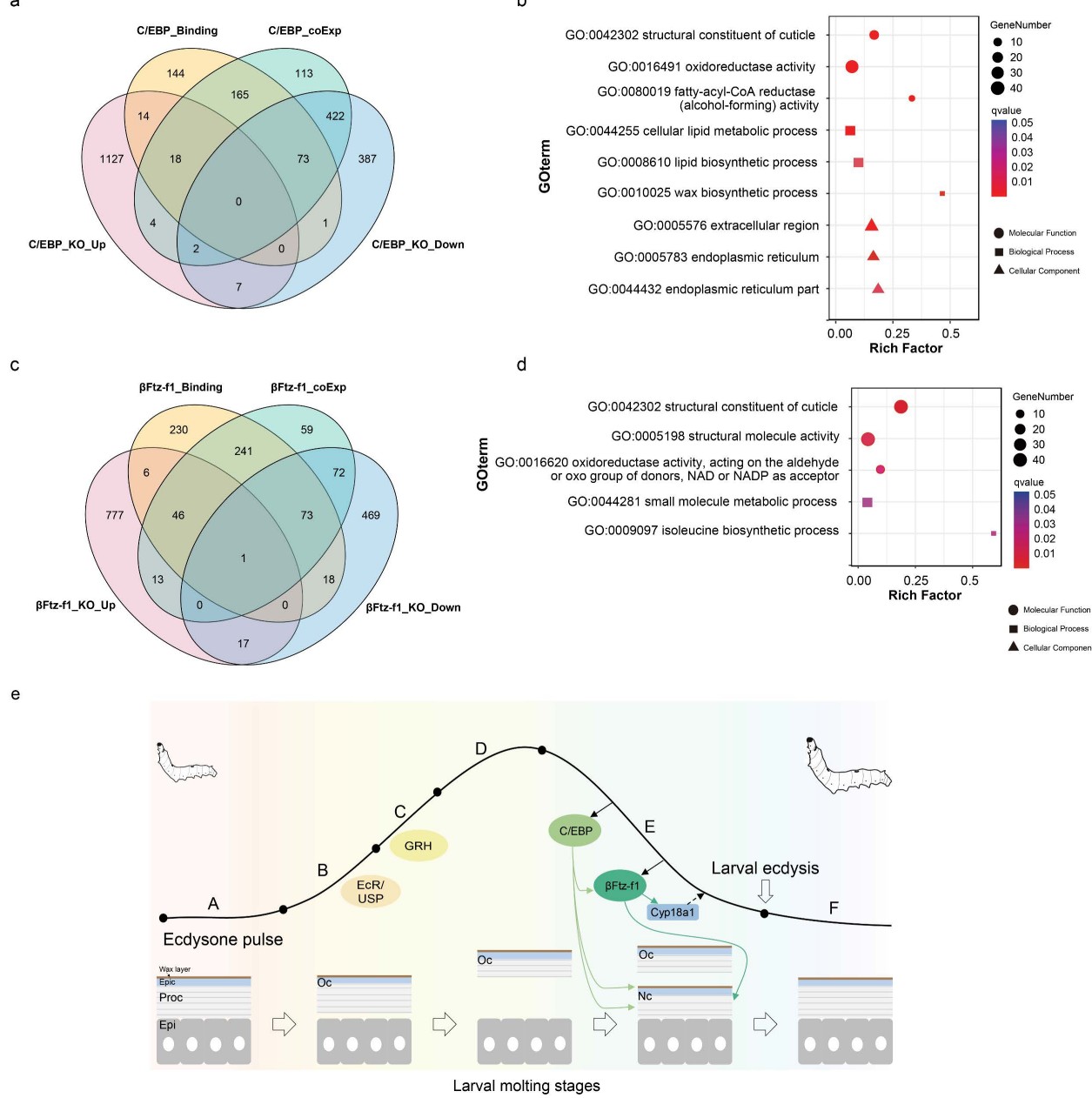

**Fig 7. Transcriptome analysis of epidermis from *C/EBP* and *βFtz-f1* mutated larvae. (a)** Venn diagram of DEGs in *C/EBP* mutated larvae, C/EBP binding genes from ATAC-seq data and *C/EBP* co-expression genes from time-series RNA-seq data. **(b)** GO terms enrichment analysis of overlay genes for C/EBP. The x-axis indicated the rich factor which which quantifies the degree of enrichment of a specific GO term within DEGs. **(c)** Venn diagram of DEGs in *βFtz-f1* mutated larvae, βFtz-f1 binding genes from ATAC-seq data and *βFtz-f1* co-expression genes from time-series RNA-seq data. **(d)** GO terms enrichment analysis of overlay genes for βFtz-f1. **(e)** A working model to explain how 20E pulse regulate the larval-larval molting process in the silkworm.

indicating critical roles of the TF in formation of new cuticle. Like other insects, βFtz-f1 expressed at late molting stages when the concentration of 20E declined to low level [17]. As a key factor, it's interesting to dissect how animal provokes *βFtz-f1* expression. In *Drosophila*, HR3 could induce *βFtz-f1* expression directly [46]. However, consistent with *Manduca*

*sexta,* silkworm *HR3* gene started to express as the 20E rises, and the signal declined when the hormone titer decreased during late larval molting process (S3 Fig), suggesting HR3 was unable to participate in the expression of the *βFtz-f1* gene in the two lepidopteran insects [17,47]. Here, we found C/EBP belonging to basic leucine zipper (bZip) transcription factor family, could bind to the promoter region of *βFtz-f1* gene. Besides, disruption of *C/EBP* in silkworm epidermal cells significantly repressed the transcription of *βFtz-f1* gene. More importantly, activation of *C/EBP* gene also required the decline of 20E level. It began to express after the decline of 20E, and reached the peak at about E1 stage. The expression timing of *C/EBP* gene was earlier than *βFtz-f1* (Fig 6A). Therefore, we speculate that C/EBP is a direct regulator to regulate the transcription of *βFtz-f1*.

In *Drosophila,* slow border cells (slbo), the silkworm C/EBP ortholog, was demonstrated to mediate the ovarian border cell migration [48]. During late embryogenesis in *Drosophila, slbo* was highly expressed in the nuclei of gut and epidermis. *Slbo* deletion mutants caused late embryonic lethal without obviously morphological defects, and the mechanism was still unclear [49]. Recently, *slbo* in *Tribolium castaneum* was identified as a regulator controlling expression of eclosion hormone [50]. In silkworm, most of the *C/EBP* mutant larvae could not synthesize normal new cuticle, fail to shed the old cuticle and died during the molting stages. Transcriptomic data from *C/EBP* mutant confirmed C/EBP could regulate the expression of cuticular protein genes, indicating the important function of the factor in epidermal development. Besides, we also found *C/EBP* took part in the deposition of the outer wax layer of the silkworm exoskeleton. Several *C/EBP* mutant larvae could finish the larval molting process, but died with the shrunk body during the feeding stage. Some wax synthesized enzymes, such as fatty acid reductases, were repressed in the mutant. Though there has no other studies about roles of insect *C/EBP* in epidermis, the members belonging to C/EBP subfamily in mammal are required for the differentiation of skin [51,52]. Therefore, the epidermal regulatory function of *C/EBP* may be conserved across animals.

Beyond the regulators identified for the late molting stage, ECR/USP and GRH are considered as two main factors controlling gene expression during early molting stages. The ECR/USP heterodimer acts as a primary regulator initiating the 20E signaling pathway. GRH is highly conserved across many metazoans and essential for the expression of many genes involved in epithelial cell fate and wound healing [53,54]. Recent studies in *Drosophila* further identified GRH as a pioneer factor regulating of epithelial enhancer accessibility [55,56]. In this study, the enrichment of GRH-binding motifs within C1-accessible chromatin regions further suggests a potential regulatory role for GRH during the molting process. However, direct molecular evidence supporting this function remains lacking in lepidopteran insects, and further investigation are needed to clarify its role in molting regulation.

In conclusion, we examined the dynamic expression pattern and chromatin status of genes during silkworm larval molting stages, and confirmed that the timing of gene expression along with the 20E titer was critical for the proper developmental events. The hormone exerts its role by activating different transcription factors in concentration manner. At the early molting stage, the increasing 20E level triggers EcR and GRH expression to promote the apolysis of the old cuticle layer. As the hormone starts to decline from peak, it first activates C/EBP controlling the deposition of procuticle layer and wax layer. More importantly, the TF can serve as a positive regulator to stimulate late 20E response factor, *βFtz-f1* which is indispensable for the transcription of bulks of cuticular protein genes (Fig 7G). Besides, *βFtz-f1* also promotes expression of *Cyp18a1,* a 20E degradation enzyme gene, which further decreases the hormone concentration. The high conservation of those transcription factors among different insects indicates their conserved regulatory function in epidermis.

## Materials and methods

### Silkworm strains

Non-diapausing silkworm strain, *Nistari,* was used for all experiments shown in this paper. The epidermal specifically driven (G25p) *Cas9* transgenic silkworm were described in our previous study and kept in our laboratory [30]. Larvae were reared on fresh mulberry leaves at 25°C under a 12 h-light: 12 h-dark photoperiod.

## Histological analysis, H&E and chitin staining

During the silkworm larval-larval molting process, Kiguchi and Agui (1981) found the characteristic of spiracle (spiracle index) could precisely distinguish different molting events corresponding to 20E fluctuation [6] (Fig 1A). C1 is a start stage of the fourth molting when the hormone is increasing, and a triangular shaped area on the dorsal side of the spiracle could be detected. The ecdysteroid titer reaches the peak at the D3 stage of the 4th instar larvae, and the newly formed cuticle around the old spiracle is first recognizable at this stage. At E1 stage, a thin layer of exocuticle is secreted and the hormone concentration is decreasing. By stage E2, the new peritreme around spiracle turns dark. Using this method, we carefully selected silkworm larvae at different molting stages, and dissected the dorsal part of abdominal integument. The collected samples then were fixed in 4% paraformaldehyde, and embedded in paraffin and cut to thick sections (6–7 μm). The sections were further dewaxed and rehydrated. Finally, the sections were stained by haematoxylin and eosin (H&E) or Fluorescent Brightener 28 (FB28) (Solarbio, China) to detect chitin.

## 20E treatment and quantification of the hormone

To examine the effect of decreasing 20E concentration on the molting process, the molting larvae at D3 stage when the hormone reached peak were chosen. DMSO (Solarbio, China), 20E (Sangon Biotech, China) (5 μg/larva) or methoxyfenozide (MedChemExpress, China) (1 μg/larva) was injected into the larval haemolymph. Then the epidermis was dissected from 3-5 treated larvae and stored in liquid nitrogen. For DMSO and 20E treated samples, haemolymph collected form 3–5 larvae was mixed with methanol to extract ecdysteroids. Ecdysteroid levels were quantified via competitive enzyme-linked immunoassay kit with anti-20E rabbit antiserum (Cayman Chemicals, USA). The method was based on the manufacturer's instruction.

## RNA and ATAC sequencing

As shown above, epidermis from different 4th molting stages or treatment were dissected. The fresh dorsal integument of the segments 6 was dissected from 10-15 larvae. Then, under a stereomicroscope, attached fat body and muscle were carefully removed by forceps from epidermis. Moreover, for C/EBP and βFtz-f1 mutated larvae, we dissected epidermis of individuals at E1 and E2 stage of 1st larval molting stages. The G25p driven Cas9 transgenic silkworm was used as the control for mutant. The total RNA of those samples was extracted by a TransZol up Plus RNA Kit according to the manufacturer's protocol (Beijing TransGen Biotech, Beijing, China). Each epidermis sample was collected from five larvae. Three biological replicates were performed for each condition. Then, the RNA was sent to the Biomarker Technologies Corporation (Beijing, China) to check the purity and integrity of RNA, and construct the cDNA library. RNA purity was checked using a Nano Photometer spectrophotometer (Implen, Westlake Village, CA, USA). RNA 6000 Nano Assay Kit and Bioanalyzer 2100 (Agilent Technologies, Santa Clara, CA, USA) were implemented to assess RNA integrity. RNA sequencing was performed by Illumina HiSeq X Ten (Illumina, San Diego, CA, United States) with 125 bp paired-end reads according to the manufacturer's instructions.

The larvae at C1 and E1 molting stages were selected to perform ATAC sequencing. The tissues were homogenized and divided to two parts. One was used to do RNA sequencing, and the other part was disassociated into cell suspensions to perform ATAC sequencing. Then the cells were lysed with cold lysis buffer to collect nuclei. Tn5 transposase was used to perform tagmentation of nuclei. The cut DNA fragments were amplified by PCR and then sequenced by second-generation high-throughput sequencing on Illumina HiSeq PE150 sequencer.

## Sequencing data processing

**RNA-seq.** Paired-end reads were aligned to the silkworm reference genome by HISAT2 with default parameters [57,58]. The mapped reads were counted by HTSeq-count [65]. DESeq2 package was used to identify the DEGs between

different groups with greater than 2-fold changes at corrected $p$-values < 0.05 [59,60]. To identify genes with dynamic temporal expression profiles over molting process, we run Next-maSigPro software [20]. By employing the R package Weighted gene co-expression network analysis (WGCNA), we further constructed co-expression networks from genes in one cluster identified by maSigPro [28].

**ATAC-seq.** After removing adaptors and low-quality reads by fastp, paired-end ATAC-seq reads were aligned to the silkworm reference genome using Bowtie2 with default parameters [61]. Mitochondrial DNA and PCR duplicates were filtered by removeChrom (https://github.com/harvardinformatics/ATAC-seq/blob/master/atacseq/removeChrom.py) and Picard program (https://broadinstitute.github.io/picard/), respectively. We further used HMMRATAC, a peak calling program specific to ATAC-seq data, to obtain the signal peak of the open area, and the peaks were annotated and assigned to different genomic regions such as transcription start site (TSS), exon, intron and untranslated region (UTR) by the R package ChIPseeker [62,63]. Then, differentially accessible regions (DARs) between the two time points were identified using *csaw* program with the parameters: loess-based normalization for removing trended biases, FDR < 0.05 [64]. To analyze potential transcription factor binding sites in the chromatin accessible regions, HOMER software by hypergeometric model was used to perform motif enrichment relative to genomic background with default parameters (http://homer.ucsd.edu/homer/). Footprint analysis was implemented by TOBIAS in conjunction with insect transcription factor motifs from JASPAR [65].

**Joint analysis of RNA-seq and ATAC-seq.** According to the ChIPseeker data, we assigned the ATAC-seq peaks located in the body of the transcript, together with 2-kb regions upstream of the TSS and downstream of the 3′ end, to the gene. If a peak was found in the overlap of two genes, one of the genes was randomly chosen. Then the DAR data and DEGs list were combined together to examine their correlation.

## Enrichment analysis

GO annotation and enrichment were performed by interproscan software and clusterProfiler, respectively [66,67]. KEGG annotation and enrichment were performed by kobas3 [68].

## Germline transformation

To knock out *C/EBP* and *βFtz-f1* genes identified in footprint analysis of ATAC sequencing, we designed two sgRNAs specifically targeting the coding sequence of each gene by the online tool (CRISPOR) [69]. Then, the sgRNAs of *C/EBP* and *βFtz-f1* genes driven by *U6* promoter was cloned into *piggyBac-IE1-Dsred-sv40* plasmid, respectively. The transgenic plasmids, helper plasmid (pHA3PIG) and *piggyBac* transposase mRNA were mixed together at final concentrations of 250 ng/μL, 200 ng/μL and 200 ng/μL, respectively. The mixture was injected into pre-blastoderm G0 embryos, and eggs were incubated at 25°C in a humidified chamber until hatching. The G0 larvae were reared into moths under standard conditions. G1 progeny were obtained by mating G0 moths with wild type moths for screening positive individuals (U6-*C/EBP*-sgRNAs; U6-*βFtz-f1*-sgRNAs) through the red fluorescence marker.

## Construction of mutant

Epidermis specific Cas9 transgenic silkworm (*G25p-Cas9*) established in our previous study was used to cross with U6-*C/EBP*-sgRNAs or U6-*βFtz-f1*-sgRNAs silkworm. Their progeny with double fluorescent (EGFP and RFP) was considered as mutant. To confirm the mutagenesis of the two genes, genomic DNA from mutant larvae were extracted, and sgRNA targeting regions were amplified and sequenced. The individuals only with EGFP signaling were used as control.

## Transmission Electron Microscope, TEM

The dorsal part of abdominal integument of newly-ecdysed 2nd instar mutated larvae and control were dissected. The samples were first fixed with 2.5% glutaraldehyde for 48 hours at 4°C, and washed three times in phosphate buffer. Then, the

samples were post-fixed in 1% osmium tetroxide for 3 h at 4°C, and rinsed three times in phosphate buffer. After dehydrating with a series of ascending concentrations of acetone, the samples were embedded in Epon 812 resin for 2 h at room temperature. Ultra-thin sections (50–60 nm) were prepared and counterstained with 3% uranyl acetate and lead citrate. The images were captured with J JEM-1400 transmission electron microscope (TEM, JEOL).

### Quantification of Chitin content

The integument of mutated and control larvae was dissected, and dried under 60°C. The same weight of samples was used to extract chitin. The method to extract and quantify the chitin was described in previous studies [27,70]. Four biological replicates for each mutated and control group.

### Mutation frequency analysis by amplicons sequencing

The integument of mutated (larvae with double fluorescents) and control (larvae with green fluorescent) larvae was dissected and cut into small pieces. Then the sample was lysis by One Step Mouse Genotyping Kit (TOLOBIO, Shanghai, China) and used as template to perform PCR amplification according to the manufacturer's instructions. The specific primers containing adaptors were listed in S1 Table. Amplicons were sent to Sangon Biotech Corporation (Shanghai, China) to construct sequencing library and were PE150 (*βFtz-f1*) or PE300 (*C/EBP*) sequenced by the company. Mutations induced around the targeted sites were analyzed with CRISPR RGEN Tools Cas-Analyzer software [71].

### Quantitative PCR

Total RNAs of samples were extracted as shown above. One microgram of the RNA was used to synthesize cDNAs by EasyScript One-Step gDNA Removal and cDNA Synthesis Kit (TransGen Biotech, Beijing, China). qRT-PCR was performed by QuantiTect SYBR Green PCR Kit (Qiagen, German) according to the manufacturer's instructions. Each sample had at least three biological repeats.

### Chromatin Immunoprecipitation (ChIP)

To confirm the binding of the transcription factors to the regulatory region of their targeted genes, the ChIP assays were performed (Beyotime, Nantong, China). The embryonic derived cell line *BmE* was cultured in Grace's insect medium (Gibco, USA) with 10% fetal bovine serum (FBS) at 27°C. After seeding in 24-well culture plates for 12 hours, the cells were transfected with 1 µg *pIZ-C/EBP-Flag* or *pIZ-βFtz-f1-Flag* vector. Eight hours later, the transfection mixture was replaced with fresh insect medium and 10% FBS. Meanwhile, 20E at final concentration of 1 µM was added to the medium. The cells were treated for 24 hours, and then the medium was replaced with fresh insect medium and 10% FBS to remove hormone. After culturing for an additional 6 hours, the cells were harvested for ChIP experiment. The cells were fixed by 1% formaldehyde for 10 min at 37°C. Then the DNA-protein complexes were sheared by sonication. Immunoprecipitation assays were performed using anti-Flag (M20008, Abmart, China) and normal mouse IgG antibodies. The purified DNA from the immunoprecipitated chromatin was used as the template for quantitative PCR. All primers used in this study are listed in S1 Table.

## Statistical analyses

All statistical analyses were performed by Student's t test, Wilcoxon rank-sum test or Mann-Whitney U test using R.

## Supporting information

**S1 Fig. Chitin staining of epidermis during the silkworm larval-larval molting stages.**
(S1_Fig.PDF)

**S2 Fig. Expression profiles of chitin metabolic genes and cuticular protein genes.** (a) Heatmap of chitin degradation (left panel) and biosynthesis (right panel) related genes. Different enzyme families were distinguished by color. (b) Heatmap of fatty-acyl-CoA reductases (FAR). (c) Heatmap of cuticular protein genes. (d) Numbers of cuticular protein genes belonging to different families were identified in specific expression cluster.
(S2_Fig.PDF)

**S3 Fig. Expression heatmap of transcription factors involving in 20E signaling pathway.**
(S3_Fig.PDF)

**S4 Fig. The concentration of 20E after exogenous hormone treatment.** The larvae at D3 stage were injected with exogenous 20E or DMSO. The 20E titer was measured from the hemolymph collected at different time points. $n = 3$ biological replicates.
(S4_Fig.PDF)

**S5 Fig. The feature of ATAC and RNA sequencing signals.** (a) Distribution of ATAC sequencing signals in the TSS of genes. (b) Genome browser shot of normalized ATAC and RNA sequencing counts of several represent genes. ATAC-seq and RNA-seq data from C1 stage in blue and data from E1 stage in red. Height indicates normalized ATAC-seq or RNA-seq signal.
(S5_Fig.PDF)

**S6 Fig. Expression profile of genes with binding sites of the transcription factors identified by ATAC-seq.** (a) Enrichment of TF motifs in ATAC-seq peak regions at C1 molting stage. "% of Targets" (y-axis) and "% of Background" (x-axis) represent the prevalence of a specific TF motif (spots) within open chromatin regions at C1 stage and genomic background, respectively. A statistically significant enrichment occurs when the "% of Targets" is significantly higher than the "% of Background". (b) Enrichment of TF motifs in ATAC-seq peak regions at E1 molting stage. "% of Targets" (y-axis) and "% of Background" (x-axis) represent the prevalence of a specific TF motif (spots) within open chromatin regions at E1 stage and genomic background, respectively. (c) Heatmap of genes with ECR/USP binding sites. (d) Heatmap of genes with GRH binding sites. (e) Heatmap of genes with C/EBP binding sites. (d) Heatmap of genes with βFtz-f1 binding sites. (f) Quantitative real-time PCR from the ChIP assays in *BmE* cells that overexpressed Flag-tagged *C/EBP*. (g) Quantitative real-time PCR from the ChIP assays in *BmE* cells that overexpressed Flag-tagged *βFtz-f1*.
(S6_Fig.PDF)

**S7 Fig. Schematic diagram for knocking out interested genes in silkworms by conditional knockout system.** (a) The U6 driven sgRNAs transgenic silkworm with red fluorescent was crossed with epidermis specifically expressed *Cas9* transgenic silkworm with green fluorescent, and their offspring with double fluorescent were selected to identify the genomic DNA mutagenesis. (b) the temporal expression pattern of *Cas9* gene in *Cas9* transgenic silkworm. The eggs and epidermis of 4th instar larvae were collected at indicated time points. Three biological replicates. (c) Mutant nucleotide sites of the *C/EBP* (upper panel) and *βFtz-f1* (bottom panel) mutated larvae. The names with -egg means the sequences were amplified by genomic DNA extract from the egg. The names with -ref means sequence from silkworm reference genome, Other sequences were amplified by the epidermis from indicated mutated larvae as templates. (d) Mutation frequency of the targeting sites by amplicons sequencing. DF means the C/EBP or *βFtz-f1* amplicons from the larva with double fluorescents. G25p means the C/EBP or *βFtz-f1* amplicons from the larva with green fluorescent as wild control.
(S7_Fig.PDF)

**S8 Fig. Differentially expressed genes identified in *C/EBP* and *βFtz-f1* mutated larvae.** (a) Heatmap of differentially expressed cuticular protein genes identified in *C/EBP* and *βFtz-f1* mutated larvae. (b) Heatmap of differentially expressed fatty-acyl-CoA reductase genes identified in *C/EBP* and *βFtz-f1* mutated larvae. (c) The relative expression level of *βFtz-f1*

(left panel) and *Cyp18a1* (right panel) gene in *C/EBP* and *βFtz-f1* mutated larvae at 1ˢᵗ larval molting E1 stage. $n = 3$ biologically replicates. **$P < 0.01$.
(S8_Fig.PDF)

**S1 Table. List of nucleotide sequence of primers used in this study.**
(S1_Table.DOCX)

## Acknowledgments

We thank Judith H. Willis for critical reading of the manuscript. We also thank all other members of Zhang's lab for their laboratory assistance.

## Author contributions

**Conceptualization:** Liang Qiao, Ze Zhang, Wei Sun.

**Data curation:** Yun Wang, Xin Yang, Junfeng Hong.

**Funding acquisition:** Liang Qiao, Wei Sun.

**Investigation:** Yun Wang, Xin Yang, Wei Sun.

**Methodology:** Yun Wang, Xin Yang, Junfeng Hong, Lingyi Li, Xia Ling.

**Resources:** Ze Zhang, Wei Sun.

**Supervision:** Ze Zhang, Wei Sun.

**Visualization:** Yun Wang, Xin Yang, Lingyi Li, Liang Qiao, Wei Sun.

**Writing – original draft:** Yun Wang, Wei Sun.

**Writing – review & editing:** Liang Qiao, Ze Zhang, Wei Sun.

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
