## [Decision Letter · Decision Letter 0]

15 May 2025

PGENETICS-D-25-00373

Transcriptomic landscape and chromatin accessibility uncover pivotal regulators driving programmed larval-larval molting in the domesticated silkworm

PLOS Genetics

Dear Dr. Sun,

Thank you for submitting your manuscript to PLOS Genetics. After careful consideration, we feel that it has merit but does not fully meet PLOS Genetics's publication criteria as it currently stands. Therefore, we invite you to submit a revised version of the manuscript that addresses the points raised during the review process.

Please submit your revised manuscript within 60 days Jul 14 2025 11:59PM. If you will need more time than this to complete your revisions, please reply to this message or contact the journal office at plosgenetics@plos.org. Please include the following items when submitting your revised manuscript:

We look forward to receiving your revised manuscript.

Kind regards,

Lynn M Riddiford

Academic Editor

PLOS Genetics

Marnie Blewitt

Section Editor

PLOS Genetics

Aimée Dudley

Editor-in-Chief

PLOS Genetics

Anne Goriely

Editor-in-Chief

PLOS Genetics

**Additional Editor Comments:**

In revising this manuscript, please pay particular attention to the comments of Reviewer 2.

**Journal Requirements:**

At this stage, the following Authors/Authors require contributions: Yun Wang, Xin Yang, Junfeng Hong, Lingyi Li, Xia Ling, Liang Qiao, Ze Zhang, and Wei Sun. Please ensure that the full contributions of each author are acknowledged in the "Add/Edit/Remove Authors" section of our submission form.

The list of CRediT author contributions may be found here: https://journals.plos.org/plosgenetics/s/authorship#loc-author-contributions

https://journals.plos.org/plosgenetics/s/submission-guidelines#loc-parts-of-a-submission

4) We noticed that you used the phrase 'data not shown' in the manuscript. We do not allow these references, as the PLOS data access policy requires that all data be either published with the manuscript or made available in a publicly accessible database. Please amend the supplementary material to include the referenced data or remove the references.

5) Please upload all main figures as separate Figure files in .tif or .eps format. For more information about how to convert and format your figure files please see our guidelines: 

6) Please ensure that the funders and grant numbers match between the Financial Disclosure field and the Funding Information tab in your submission form. Note that the funders must be provided in the same order in both places as well. State what role the funders took in the study. If the funders had no role in your study, please state: "The funders had no role in study design, data collection and analysis, decision to publish, or preparation of the manuscript.".

**Reviewers' comments:**

Reviewer's Responses to Questions

**Comments to the Authors:**

Reviewer #1: In this manuscript, authors explain the results of transcriptome analysis comparing with morphological changes of the epidermis. They divided the stage into early, middle and late molting stages. Then, they divided genes into six clusters depending on the expression pattern. They examined the effect of 20E treatment. They operated ATAC-seq and ANA-seq analysis. Through the several bioinformatical methods, they found ECR/USP and Grainy head (GRH) motifs are two most prominently enriched in C1, and C/EBP and βFtz-f1 in E1. They examined the effects after treatment of hormone, the effect of KO of C/EBP and βFtz-f1, and speculated that C/EBP is a direct regulator to initiate the transcription of βFtz-f1. The information presented is valuable to readers, they found two prominent transcription factors in the early molting stage, and late molting stage respectively. However, the following points should be addressed in the revision:

The consideration on the function of GRH is not enough. The reason of GRH motif enrichment and the possibility of its function should be more deeply discussed. They speculated that C/EBP is a direct regulator to initiate the transcription of βFtz-f1, but they did not show direct interaction of two, so their speculation should be changed. Explanation of Figs is too simple. I think authors should explain every figures more clealy.

Minor comments are below.

Explanation of the number, C1, D3, E1, E2 is required.

Fig. 1a: illustrated ecdysteroid titer is required.

Fig. 1c: More explanation is need, PC1, PC2, PC3 are difficult to understand, and in the present, I think this figure should be deleted or more explanation need.

Fig. 2b, BP, CC, MF; explanation is required here not in Fig. 7.

Fig. 2d, 2e: More explanation is required.

Fig. 2f, Explanation of colors and numbers are required.

Fig. 5a, b; More explanation is need.

Fig. 7b, d: rich factor?

L148: this stage, hormone increase?

L189~: 20E titer in the hemolymph is ecdysteroid titer, not 20E.

L420; that, instead of the.

Reviewer #2: In this manuscript, the authors investigate gene expression dynamics and chromatin accessibility changes during the larval-larval molt in the silkworm Bombyx mori, using transcriptomic and ATAC-seq analyses.

While the methodological development in a non-model insect species is appreciated, much of the content looks descriptive.

It remains unclear to me whether the large-scale datasets generated in this study yield truly novel or impactful biological insights.

Major Comments:

1. Clarify novel versus known findings

The manuscript does not clearly distinguish between findings that are newly uncovered in this study and those that are reconfirmations of previously reported results.

These two categories should be clearly separated.

For example, Fig. 1a and 1b do not present novel findings.

Similarly, Fig. 2f reproduces a known result.

The authors should cite the relevant previous studies and clearly state that their data confirm these findings, thereby supporting the technical validity of their transcriptomic analysis.

2. GRH-related claims in the abstract

The authors describe GRH as a key early-stage regulator in the abstract, but no functional characterization of GRH is provided in the manuscript.

This statement appears to be an overinterpretation and should be toned down.

3. Use of the CPG25 promoter to drive Cas9

The authors use the CPG25 promoter to express Cas9, but this promoter is known to be active during the late E2 stage (Wang et al. 2022, IBMB).

Since C/EBP is expressed earlier, and bFTZ-F1 peaks around E1–E2, it is questionable whether this system is appropriate for analyzing these genes.

Moreover, the authors did not generate a knockout line or perform G0 somatic mosaic analysis—what was the reason for this choice?

Although induced mutations are reported, they should be treated quantitatively.

For example, the authors could separately extract genomic DNA from epidermis and non-target tissues (e.g., gut or fat body) and perform amplicon sequencing to assess mutation frequency.

4. Epidermis dissection protocol

The dissection of the epidermis is not well described.

The authors should specify which body segments were used and clarify how thoroughly fat body and muscles were removed during tissue collection.

5. Discussion section is overly long

The Discussion is disproportionately long and at times redundant.

The authors are encouraged to streamline the section to focus on key messages and reduce unnecessary elaboration.

Minor Comments:

Fig. 1b: "epidermis cells" → "epidermal cells"

Line 126: "time serial transcriptome" → "time-series transcriptome" or "time-course transcriptome"

Line 131: "B to D3" → possibly "C1 to D3"?

Line 148: "they were the hormone inducible" → "they were 20E-inducible genes"

Line 176: The use of "more" is unclear, as it is not specified what factors are being compared.

Fig. 1d: What does "Knickkopf" refer to here?

Line 189: "Hormone regulation of molting process" → "Hormonal regulation of the molting process"

Line 335: "time-serial" → "time-series"

Reviewer #3: This MS contains some interesting results on the regulation of larval molting in the silkworm. Integrated analysis of RNA seq and ATAC seq data identifies new transcription factors involved in the regulation of expression of cuticular protein genes. Overall, the research is well designed, and appropriate approaches were employed, and results were interpreted well, and the conclusions are supported by the data included in the MS. Suggestions below are to help improve the MS.

1. The authors appear to use “molting” and “ecdysis" interchangeably, as well as “mRNA levels” and “gene expression”; please correct the manuscript to ensure proper usage.

2. Please define abbreviations (e.g. EcR, GRH, C/EBP) used in the abstract and at first appearance in the text.

3.The Figures are crowded and therefore hard to get to the important point. I recommend moving some of the supporting Figures to supplementary data.

4. Line 51 frameworks are

5. Line 56, Endocrine mechanisms, hormones?

6. Line 148, indicating that they are hormone inducible

7. Line 138, genes with typical expression

8. Line 197, As the hormone titers begin to increase (B stage), the transcript levels of ecdysone receptor (EcR) and its partner ultraspiracle (USP) also increased.

9. Line 239, binding sites. Epidermis from two time points (Fig. 1a, C1 and E1), representing early and late molting stages, was selected for ATAC-seq

10. Line 246, showed that genes with high expression levels are distributed around the open chromatin

11. Line 265, the expression profiles of the transcription factors were positively correlated with those of their target genes

12. Line, 271, E1 stage, their targeted genes were enriched in new epidermal synthesis processes, such as structural constituents of cuticle, chitin binding, lipid biosynthetic processes

13. Line 281, The two genes started to transcript from E1 stage indicating ecdysone-not clear, rewrite.

14. Line 285, the hormone titers decreased to a level

15. Line 329, essential to the formation of new cuticle

16. Line 358 and throughout MS, replace Firstly with first.

17. Line 387, CPs from various families showed high and specific expression pattern during the late molting stages.

18. Please state the sources of chemicals and reagents used.

19. Line 519, sequencing. Then the cells were lysed with cold lysis buffer to collect nuclei. Tn5 transposase was used to perform tagmentation of nuclei.

**Have all data underlying the figures and results presented in the manuscript been provided?**

Reviewer #1: None

Reviewer #2: Yes

Reviewer #3: Yes

PLOS authors have the option to publish the peer review history of their article (what does this mean? ). If published, this will include your full peer review and any attached files.

**Do you want your identity to be public for this peer review?** For information about this choice, including consent withdrawal, please see our Privacy Policy .

Reviewer #1: No

Reviewer #2: No

Reviewer #3: No

**Figure resubmission:**
---

## [Decision Letter · Decision Letter 1]

18 Jul 2025

PGENETICS-D-25-00373R1

Transcriptomic landscape and chromatin accessibility uncover pivotal regulators driving programmed larval-larval molting in the domesticated silkworm

PLOS Genetics

Dear Dr. Sun,

Thank you for submitting your manuscript to PLOS Genetics. After careful consideration, we feel that it has merit but does not fully meet PLOS Genetics's publication criteria as it currently stands. Therefore, we invite you to submit a revised version of the manuscript that addresses the points raised during the review process.

Please submit your revised manuscript within 30 days Aug 17 2025 11:59PM. If you will need more time than this to complete your revisions, please reply to this message or contact the journal office at plosgenetics@plos.org. Please include the following items when submitting your revised manuscript:

We look forward to receiving your revised manuscript.

Kind regards,

Lynn M Riddiford

Academic Editor

PLOS Genetics

Marnie Blewitt

Section Editor

PLOS Genetics

Aimée Dudley

Editor-in-Chief

PLOS Genetics

Anne Goriely

Editor-in-Chief

PLOS Genetics

**Additional Editor Comments (if provided):**

Please consider the comments of Reviewer 2 in making your revision.

**Journal Requirements:**

Please ensure that the CRediT author contributions listed for every co-author are completed accurately and in full.

At this stage, the following Authors/Authors require contributions: Yun Wang, Xin Yang, Junfeng Hong, Lingyi Li, Xia Ling, Liang Qiao, Ze Zhang, and Wei Sun. Please ensure that the full contributions of each author are acknowledged in the "Add/Edit/Remove Authors" section of our submission form.

The list of CRediT author contributions may be found here: https://journals.plos.org/plosgenetics/s/authorship#loc-author-contributions

**Reviewers' comments:**

Reviewer's Responses to Questions

**Comments to the Authors:**

Reviewer #1: Recommendation: Accept

Reviewer #2: The authors have generally addressed my previous comments appropriately in this revision. However, there is still one point that remains unaddressed.

Major Comments:

1. Analysis of the outcome of the transgenic CRISPR system

Previously, I requested a quantitative analysis of the mutant alleles generated by the authors' transgenic CRISPR system.

In the current revision, the authors only present the types of mutant alleles, but do not provide any data showing what proportion of the amplicons carry those mutations.

In an experimental design like this, where F1 individuals derived from crossing Cas9-expressing and sgRNA-expressing lines are analyzed, rather than fixed mutant lines, I think it is especially important to assess the frequency of mutant alleles quantitatively.

Whether the frequency is close to 100% or at an intermediate level (which may result in mosaic phenotypes) is crucial for evaluating the efficiency and reliability of the transgenic CRISPR system.

I strongly encourage the authors to perform amplicon sequencing using epidermal samples from both Cas9-sgRNA individuals and appropriate controls (e.g., sgRNA-only individuals), and to analyze the data quantitatively.

Importantly, the authors have already dissected the epidermis of L1 larvae produced via the transgenic CRISPR system and used them for RNA-seq and ATAC-seq analyses (Figs. 7 and S8).

Therefore, epidermis dissection in L1 is clearly feasible for the authors, and conducting amplicon-seq on those samples should be straightforward.

Finally, the raw amplicon-seq data should be deposited in a public repository.

Minor Comments:

1. The short title is somewhat vague, as it does not clearly indicate what regulates the cuticle.

Clarifying the main subject of the study would help readers quickly grasp the focus of the work.

Reviewer #3: I am satisfied with the revisions performed.

**Have all data underlying the figures and results presented in the manuscript been provided?**

Reviewer #1: None

Reviewer #2: Yes

Reviewer #3: Yes

PLOS authors have the option to publish the peer review history of their article (what does this mean? ). If published, this will include your full peer review and any attached files.

**Do you want your identity to be public for this peer review?** For information about this choice, including consent withdrawal, please see our Privacy Policy .

Reviewer #1: No

Reviewer #2: No

Reviewer #3: No

**Figure resubmission:**
---

## [Editor Report · Decision Letter 2]

11 Aug 2025

Dear Dr Sun,

We are pleased to inform you that your manuscript entitled "Transcriptomic landscape and chromatin accessibility uncover pivotal regulators driving programmed larval-larval molting in the domesticated silkworm" has been editorially accepted for publication in PLOS Genetics. Congratulations!

Yours sincerely,

Lynn M Riddiford

Academic Editor

PLOS Genetics

Marnie Blewitt

Section Editor

PLOS Genetics

Aimée Dudley

Editor-in-Chief

PLOS Genetics

Anne Goriely

Editor-in-Chief

PLOS Genetics

Comments from the reviewers (if applicable):

Thank you for your revisions. This paper is now ready for publication.

**Data Deposition**

http://datadryad.org/submit?journalID=pgenetics&manu=PGENETICS-D-25-00373R2

**Press Queries**

---

## [Editor Report · Acceptance letter]

PGENETICS-D-25-00373R2

Transcriptomic landscape and chromatin accessibility uncover pivotal regulators driving programmed larval-larval molting in the domesticated silkworm

Dear Dr Sun,

We are pleased to inform you that your manuscript entitled "Transcriptomic landscape and chromatin accessibility uncover pivotal regulators driving programmed larval-larval molting in the domesticated silkworm" has been formally accepted for publication in PLOS Genetics! Your manuscript is now with our production department and you will be notified of the publication date in due course.

With kind regards,

Anita Estes

PLOS Genetics

On behalf of:
